# Surgical Outcomes, Long-Term Survivals and Staging Systems of World Health Organization G3 Pancreatic Neuroendocrine Tumors

**DOI:** 10.3390/jcm11185253

**Published:** 2022-09-06

**Authors:** Min Yang, Lin Zeng, Shengzhong Hou, Bole Tian, Shuguang Jin, Yi Zhang

**Affiliations:** 1Department of Pediatric Surgery, West China Hospital of Sichuan University, Chengdu 610041, China; 2President & Dean’s Office, West China Hospital of Sichuan University, Chengdu 610041, China; 3Department of Pancreatic Surgery, West China Hospital of Sichuan University, Chengdu 610041, China

**Keywords:** pancreatic neuroendocrine tumors, G3, resection, stage, prognosis

## Abstract

Background: In 2017, the World Health Organization (WHO) defined a new category of pancreatic neuroendocrine neoplasms named G3 pancreatic neuroendocrine tumors (p-NETs), whose surgical outcomes, long-term survivals and staging systems have not been well documented. Methods: Data from eligible patients with G3 p-NETs defined using the WHO 2017 grading classification at our institute were retrospectively analyzed. Results: Our study enrolled 80 patients with WHO G3 p-NETs, including 50 women and 30 men. The accumulative 5-year overall survival (OS) of G3 p-NETs was 29.7%. The current staging system by the American Joint Committee on Cancer (AJCC) failed to discriminate the survival difference between Stage II and Stage III (*p* = 0.172), while notable differences with regard to the OS were statistically offered between each stage using the modified tumor–node–metastasis (mTNM) staging system (all *p* < 0.05). The OS of patients receiving surgical resection was significantly better than those with palliative operation (*p* < 0.05). Both the current AJCC system and proposed mTNM system were independent predictors for the OS of G3 p-NETs (*p* = 0.017 and *p* = 0.032, respectively). The 95% confidence intervals of the proposed mTNM staging system were smaller than that of the current AJCC system (0.626–8.217 and 0.329–10.013, respectively), indicating a relatively more accurate predictive ability. Conclusion: Our demonstration revealed that surgical resection was an independent predictor for the favorable prognosis of patients with G3 p-NETs. Moreover, the new mTNM staging system was more suitable and practical than the current AJCC system for stratifying G3 p-NETs into prognostic groups.

## 1. Introduction

Pancreatic neuroendocrine neoplasms (p-NENs) are a group of rare and highly heterogeneous tumors [1,2]. Although p-NENs were first reported in 1902 [3], the history of classifying patients into prognostic groups has experienced a long and complicated evolution, probably due to their rarity and heterogeneity [4].

In 2000, referring to some well-known clinic-pathological features, the World Health Organization (WHO) firstly classified p-NENs into well-differentiated endocrine tumor, well-differentiated endocrine carcinoma and poorly-differentiated endocrine carcinoma [5]. In 2006, based on the mitotic rate per 10 high power fields (HPFs) and Ki-67 proliferative index, the European Neuroendocrine Tumor Society (ENETS) specifically proposed a grading classification for p-NENs, which mainly consist of G1/G2 pancreatic neuroendocrine tumors (p-NETs) and G3 pancreatic neuroendocrine carcinomas (G3 p-NECs) [6]. Obtaining widespread acceptance in clinical practice, this ENETS system for p-NENs was officially adopted in 2010 by the WHO [7]. However, tumor differentiation based on morphology was not considered in the ENETS 2006 or the WHO 2010 grading classification, in which morphologically well-differentiated p-NETs might have proliferative rates that met the threshold for G3 NECs [8]. Several studies have reported that G3 NECs were also heterogeneous, which included a subgroup with clinical features close to G1/G2 p-NETs on the basis of immunostaining and gene analysis results [9,10,11,12,13]. In 2017, referring to the features of both morphological differentiation and grading upon proliferation rate, the WHO divided p-NENs into G1/G2/G3 p-NETs and G3 p-NECs [14]. In this newly updated WHO 2017 grading system, G3 p-NETs were explicitly defined as high-grade neoplasms (Ki-67 > 20%) with a well-differentiated morphology, which have not yet been well documented in terms of their surgical outcomes, long-term survivals and staging systems.

In 2017, the 8th edition staging manual by the American Joint Committee on Cancer (AJCC) first highlighted that G1/G2 p-NETs should be staged by the ENETS tumor–node–metastasis (TNM) system primarily proposed for p-NENs [6], while G3 p-NECs be staged separately by the contemporary system originally applied to pancreatic exocrine adenocarcinomas (p-EACs) [15]. Although the AJCC 8th staging manual has made an important step towards adopting uniform systems to stratify different grading p-NENs, it has lost sight of the heterogeneous features of G3 p-NECs, as we mentioned above [9,10,11,12,13]. Our previous studies identified two subgroups of G3 p-NECs with varied morphological differentiations, staging features and long-term survivals [13,16]. Meanwhile, studies have demonstrated that the current AJCC system for p-NETs failed to significantly distinguish survivals between Stage I and Stage II or between Stage II and Stage III [17,18,19,20]. Recently, Zhang et al., introduced a modified tumor–node–metastasis (mTNM) system for p-NETs [21], which adopted their previously proposed nodal classification for N definitions [22], but retained the current AJCC T and M definitions (Appendix A). This new mTNM system was proven to be superior to the current AJCC system and was highly appraised by specialists [23,24], for it offered statistically significant survival rates between each stage for G1/G2 p-NETs. Nevertheless, whether this proposed mTNM staging system is practical and prognostic for G3 p-NETs remained unknown.

In this research, we comprehensively assessed the clinical features, surgical outcomes, long-term survivals and prognostic factors of G3 p-NETs. Moreover, we attempted to validate and compare the clinical applications of this new mTNM staging system and the current AJCC system to G3 p-NETs.

## 2. Methods

Our present study was a retrospective study with regard to patients with surgically-treated and histopathologically confirmed G3 p-NETs from January 2002 to December 2020 in our hospital. Patients with a pathological diagnosis of G1/G2 p-NETs or G3 p-NECs were excluded. This study was approved by the local Institutional Review Board and Ethics Committee. In accordance with the principles of the Helsinki Declaration [25], the written informed consent was acquired on admission from all patients. The data, such as demographic baseline, clinical presentations, imaging examinations, surgical findings, pathological results and perioperative outcomes were reviewed from the patients’ medical records and collected in the prepared tabulations, as in previous work [13,16,19,20].

The surgical specimens from the tumor tissues of eligible patients were re-stained with hematoxylin–eosin and immunohistochemical methods and microscopically reviewed by our experienced pancreatic pathologists according to the morphological feature, mitotic count, Ki-67 positive proliferation index, and so on. Afterwards, G3 p-NETs were defined in the light of the WHO 2017 grading classification [14]. Meanwhile, the newly proposed mTNM staging system [21] and the current AJCC system [15] were respectively applied to distribute patients into the corresponding groups.

Follow-up was mainly conducted by telephone, email, mail or outpatient clinic review between July and December of 2021, as in previous work [13,16,19,20]. The overall survival (OS) was calculated as the time in months between the date of operation and the date of death or last follow-up, which was presented as either median survival time (MST) or 5-year OS with a hazard ratio (HR) and 95% confidence intervals (CIs). Quantitative variables were reported as median with a range, while categorical variables were presented as numbers with frequencies and proportions (%). Accumulated OS was estimated using Kaplan–Meier (K-M) methods and compared using the log-rank test. Univariate and multivariate analyses using the Cox regression proportional hazards model were performed to validate the prognostic value of potential factors for the OS of G3 p-NETs. All statistical analyses were carried out using IBM SPSS 28.0 statistical software. Differences with a two-sided *p*-value less than 0.05 were considered statistically significant.

## 3. Results

As shown in Table 1, the present study finally identified 80 eligible patients with G3 p-NETs. Our research consisted of 50 females and 30 males, with a median age of 50 years (ranging from 7 years to 74 years). There were 51 cases located in the body or tail of the pancreas, with a median diameter of 4.5 cm (ranging from 1.8 cm to 8.5 cm). Most G3 p-NETs were solitary (88.8%), and non-functional ones accounted for the majority in the tumor type (67.5%). There were 15 patients who were diagnosed as G3 p-NETs incidentally. In terms of the immunohistochemistry, the median Ki-67 proliferation index of G3 p-NETs was 28% (ranging from 21% to 60%), while the mitotic rate ranged from 20 per 10 HPFs to 38 per 10 HPFs, with a median of 22 per 10 HPFs. All G3 p-NETs had the presence of Chromogranin A (CgA) in the immunostaining. There were 12 cases of G3 p-NETs that showed the presence of necrosis. The median number of harvested lymph nodes was 8, ranging from 4 to 14.

According to the same definitions of T status by both staging systems, there were 12, 14, 32 and 22 patients classified from T1 to T4, respectively. Nodal metastasis was detected in 24 patients, including 15 cases with 1–3 regional lymph node metastases and 9 with ≥4 regional lymph node metastases, while distant metastasis was confirmed in 13 patients. In light of the corresponding clinical stages by the current AJCC 8th system, there were respectively 5, 35, 27 and 13 patients defined as Stage I, Stage II, Stage III and Stage IV. With regard to the criteria of the proposed mTNM system, there were 9, 27, 31 and 13 patients distributed from Stage I to Stage IV, respectively.

All patients were surgically treated (Table 2), of which 62 patients received surgical resection, while 18 patients received palliative operation (such as cholangiojejunostomy, gastrojejunostomy, etc.). For patients with a resection, 56 presented both grossly and microscopically negative surgical margins. The main surgical procedures performed for G3 p-NETs were distal pancreatectomy (35.0%), pancreaticoduodenectomy (21.3%) and the local resection of pancreatic tumor (13.8%), while radical resection for selected metastatic disease was only carried out in six patients. As for the anesthesia grade by the American Society of Anesthesiologists, there were respectively 9, 23, 30, 18 and 0 patients from grade I to grade V. There were 24 patients who required perioperative blood transfusions with a median volume of 300 mL, and 15 patients who needed an intensive care unit stay with a median duration of 3 d. The median duration of operation, postoperative and total in-hospital stay was respectively 150 min, 6 d and 9 d. Postoperative complications occurred in 21 patients, with a morbidity of 26.3%, in which pancreatic fistula (12.5%), intra-abdominal infection (8.8%), delayed gastric emptying (5.0%) and intra-abdominal hemorrhage (2.5%) were the main ones. One patient underwent reoperation due to intra-abdominal hemorrhage, while all other complications were treated conservatively. There was no postoperative in-hospital death. Postoperative medical therapies were carried out for 34 patients, including 14 with novel molecular targeting treatments and 20 with traditional chemotherapies.

As Table 2 listed, the median follow-up time of our study was 58.3 months (ranging from 9.7 months to 182.6 months ). When the follow-up ended, 10 patients were out of contact (12.5%). There were 39 deaths related to the disease progression (55.7%). The accumulative 5-year OS of G3 p-NETs was 29.7% (Figure 1), with an MST of 49.2 months (95% CIs: 41.8 months–56.5 months ). The accumulated 5-year OS from current AJCC Stage I to Stage IV was 100.0%, 31.3%, 17.1% and not applicable (NA), respectively (Figure 2). Patients classified using the current AJCC Stage I had better survival than those in Stage II (*p* = 0.003), Stage III (*p* = 0.006) and Stage IV (*p* < 0.001), as well as when comparing Stage II with Stage IV (*p* < 0.001) or comparing Stage III with Stage IV (*p* < 0.001). However, the survival comparison between the current AJCC Stage II and Stage III was not significant (*p* = 0.172). The 5-year OS for the proposed mTNM Stage I, Stage II, Stage III and Stage IV was respectively 100%, 39.1%, 15.6% and NA (Figure 3). Patients defined as mTNM Stage I had better survival than those at Stage III (*p* = 0.005) and Stage IV (*p* < 0.001), as well as those at Stage II compared with Stage III (*p* = 0.016) and Stage IV (*p* < 0.001). Meanwhile, the comparisons of OS between Stage I and Stage II or between Stage III and Stage IV were both statistically significant (*p* = 0.004 and *p* < 0.001, respectively).

As listed in Table 3, patients’ gender (*p* = 0.236) and age (*p* = 0.121), tumor location (*p* = 0.415), incidental diagnosis (*p* = 0.478), mitotic rate (*p* = 0.125), harvested lymph nodes (*p* = 0.512), postoperative complications (*p* = 0.517), duration of operation (*p* = 0.343) and postoperative in-hospital stay (*p* = 0.952) were demonstrated to have no notable impacts on the OS of G3 p-NETs, while the survival analyses referring to tumor type (*p* = 0.012), tumor diameter (*p* = 0.016), Ki-67 index (*p* = 0.035), necrosis (*p* = 0.027), operation classification (*p* < 0.001), postoperative medical therapy (*p* = 0.042), current AJCC 8th staging system (*p* < 0.001) and proposed mTNM staging system (*p* < 0.001) were statistically significant in univariate analyses. Using multivariate analyses in different Cox regression models, we concluded that only operation classification (*p* = 0.031 and *p* = 0.027, respectively), current AJCC 8th staging system (*p* = 0.017) and proposed mTNM staging system (*p* = 0.032) were independent predictors for the OS of G3 p-NETs. Meanwhile, the 95% CIs of the proposed mTNM staging system (0.626–8.217) were smaller than those of the current AJCC 8th staging system (0.329–10.013), indicating a relatively more accurate predictive ability.

## 4. Discussion

As we knew, G1/G2 p-NETs were regarded as well-differentiated, while G3 p-NECs were poorly-differentiated according to the grading classification by ENETS and the WHO [5,6]. However, subsequent studies revealed that, although all poorly-differentiated neuroendocrine carcinomas had a high proliferation rate, not all p-NENs with a proliferation rate above 20% were poorly-differentiated, indicating the heterogeneity of G3 p-NECs [8,9,10,11,12]. Referring to both the tumor morphology and Ki-67 index, the WHO incorporated a new subcategory of “well-differentiated high-grade tumors (i.e., G3 p-NETs)” into the well-differentiated p-NETs category in its 2017 *Classification of the Tumors of Endocrine Organs* [14], which was proven to be superior to the WHO 2010 criteria [20]. Nevertheless, the clinical features of G3 p-NETs have not yet been well documented.

In the present research, we made an in-depth analysis with regard to the surgical outcomes, prognostic factors and staging systems of G3 p-NETs. We revealed that the baseline demographics and tumor characteristics of G3 p-NETs, such as patients’ gender and age, tumor location and type, were in agreement with our previous results [16,20]. As we demonstrated in Table 3, patients with non-functional G3 p-NETs showed significantly worse survivals than those with functional tumors (*p* = 0.012), while the other factors had no obvious influence on the OS of G3 p-NETs. However, tumor type could not be a significant prognostic factor for the OS of G3 p-NETs (*p* = 0.512 and *p* = 0.214, respectively), as we previously demonstrated [16]. Moreover, while the CgA was expressed in all G3 p-NETs in the immunohistochemical examinations, we failed to test the plasma CgA values in the present study due to our limited technologies. Massironi et al., reported that plasma CgA had a significant prognostic relevance for patients with gastroenteropancreatic neuroendocrine neoplasms [26], while the prognostic value of plasma CgA for patients with G3 p-NETs still needed to be validated in future studies.

Accumulated studies reported that p-NENs at the lower end of the G3 range might, in fact, be well-differentiated with elevated Ki-67 proliferative rates and better survivals [8,9], which intrinsically prompted the formation of the WHO’s 2017 grading classification [14]. However, the role of the Ki-67 index for the new group of G3 p-NETs remains unknown due to the currently limited data in the literature. In 2018, Mizuno et al. [27] identified 10 patients with G3 p-NETs, with a median Ki-67 index of 35% (ranging from 20% to 90%), although the impact of Ki-67 on the survival of G3 p-NETs was not evaluated. Recently, de Mestier et al. [28] reported 74 patients with digestive well-differentiated G3 neuroendocrine tumors (including 53 cases located in pancreas/duodenum), with a median Ki-67 index of 30% (ranging from 21% to 80%). Meanwhile, de Mestier et al. [28] demonstrated that the Ki-67 index was not a significant predictor for the progression-free survival of these patients. In our study, the median Ki-67 index of this cohort was 28%, which was very close to the above reported data [27,28], as well as our previous results [16]. According to our validation, the Ki-67 index did indeed influence the prognosis of G3 p-NETs (*p* = 0.035), but failed to be a significant prognostic factor for the patients’ OS estimate (*p* = 0.873 and *p* = 0.435, respectively).

As reported [29,30], the molecular features and prognosis of G3 p-NETs largely differ from those of G3 p-NECs and are much closer to those of G2 p-NETs, while the most appropriate management for G3 p-NETs is currently undefined. Several studies suggested that G3 p-NETs should be treated as G2 p-NETs with respect to both surgical programs and systemic therapies [31,32,33]. Feng et al. [32] reported that the median survival was higher in patients undergoing surgery, while non-surgical management was a poor prognostic factor associated with reduced disease-specific survival in patients with G3 p-NETs. Yoshida et al. [33] revealed that surgical procedures for G3 p-NETs and G3 p-NECs should be considered separately, and that patients with G3 p-NETs could significantly benefit from surgical resection for both primary pancreatic tumors and selected metastatic disease. Meanwhile, the MST in Yoshida et al.’s research was lower than that in our report (33 months and 49.2 months, respectively), which could be explained by differences in the inclusive criteria in each cohort. What is more, we demonstrated that surgical resection was an independent and favorable predictor for the survival of G3 p-NETs (*p* = 0.031 and *p* = 0.027, respectively), which was consistent with the reports by Yoshida et al. [33]. Unfortunately, only six selected patients with metastatic disease in our study received radical resection accompanied by pancreatic surgery, making it difficult to evaluate its impact on patients’ survival.

Surgery is the optimal and curative treatment for p-NENs, while drug therapy is also very important and effective in terms of systemic treatment [1,4]. Studies have proposed that molecular targeted drugs such as sunitinib and everolimus be recommended for patients with G1/G2 p-NETs, while platinum-based chemotherapies are the first-line drugs for all p-NENs except G1/G2 p-NETs [34]. However, there have been no standardized and well-recognized medical therapeutic schedules for G3 p-NETs. Mizuno et al. [27] reported that sunitinib was as effective for G3 p-NETs as for G1/G2 p-NETs, which could significantly improve both progression-free survival and OS by reducing the tumors’ volume. Moreover, de Mestier et al. [28] revealed that adenocarcinoma-like and alkylating-based chemotherapies were the most effective treatments for advanced G3 neuroendocrine tumors regarding objective response and progression-free survival, while etoposide–platinum chemotherapy had poor efficacy in that setting. Our study enrolled 20 patients with postoperatively traditional chemotherapies and 14 patients with novel molecular targeting treatments. The changes of drug therapy for G3 p-NETs might be the result of the varied recognitions for this new subcategory of p-NENs. We demonstrated that postoperative medical therapy had notable impacts on the OS of G3 p-NETs (*p* = 0.042), although it could not be an independent predictor (*p* = 0.518 and *p* = 0.892, respectively). However, we failed to compare the impacts of traditional chemotherapies and molecular targeting treatments on the OS of G3 p-NETs due to their different drug schemes in the limited cases of this study.

The current AJCC 8th staging manual for p-NENs elucidates stratifying G1/G2 p-NETs and G3 p-NECs into different stages separately, while the most practical and appropriate staging system for G3 p-NETs remains unclear [15]. Although we previously demonstrated that G3 p-NETs might also be staged using the same AJCC system as the current one for G1/G2 p-NETs [16], this system has so far failed to distinguish prognosis among patients with Stage I vs. Stage II disease or Stage II vs. Stage III disease [17,18,19,20]. Recently, a new mTNM staging system on the basis of the current AJCC system was proposed and assessed for G1/G2 p-NETs [21], which was highly appraised [23,24], but not yet validated for G3 p-NETs. We hereby succeeded in defining G3 p-NETs into four stages using both the current AJCC staging system and the proposed mTNM approach. Furthermore, the current AJCC system failed to discriminate the survival difference between Stage II and Stage III (*p* = 0.172; Figure 2), as You et al., demonstrated [18], while notable differences with regard to the OS of G3 p-NETs were statistically offered between each mTNM stage (all *p* < 0.05; Figure 3). Meanwhile, although both systems were prognostic for predicting the OS of G3 p-NETs (*p* = 0.032 and *p* = 0.017, respectively), the 95% CIs of the mTNM staging system were smaller than that of the current AJCC system (0.626–8.217 and 0.329–10.013, respectively), indicating a potentially more accurate predictive ability. Our results of the comparisons between the applications of the mTNM system and the current AJCC approach to G3 p-NETs were similar to the validations of Zhang’s study for G1/G2 p-NETs [21], suggesting that the newly proposed mTNM staging system was more suitable and practical for G3 p-NETs.

Our study had several limitations. First of all, it was a retrospective study from a single medical institution, leading to a small number of enrolled patients with a long follow-up time. Secondly, our study excluded those patients without surgery, which meant that comparisons could not be made between the clinical features and survival differences of patients with surgical treatments and non-surgical therapies. In addition, as we mentioned above, our study failed to compare the prognosis between the resection of primary tumors and metastatic diseases, as well as between traditional chemotherapies and molecular targeting treatments, due to our limited cases. Finally, the mTNM staging system for G1/G2 p-NETs was originally designed by Zhang et al. [21], while our study for G3 p-NETs was supplementary research for the indications of this new proposed system. Therefore, a multi-center, large-volume and prospective study is still needed to confirm our results.

## 5. Conclusions

According to our in-depth analyses, tumor type, Ki-67 index, necrosis and postoperative medical therapy had certain impacts on the survival of patients with G3 p-NETs, while surgical resection was an independent and favorable predictor for patients’ OS estimate. Meanwhile, the newly proposed mTNM staging system was superior to the current AJCC system due to its better prognostic stratification and more accurate predicting ability for the OS of patients with G3 p-NETs, supporting its wider clinical use.

## Figures and Tables

**Figure 1 jcm-11-05253-f001:**
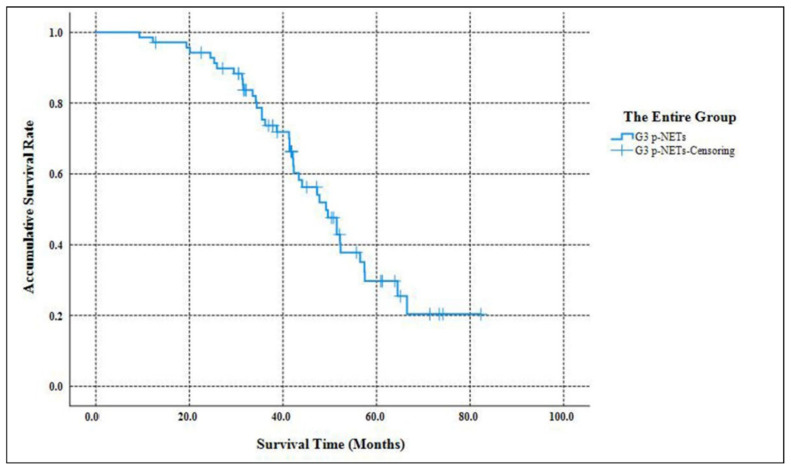
Kaplan–Meier estimates for the OS of the entire group of G3 p-NETs defined by the WHO 2017 grading classification.

**Figure 2 jcm-11-05253-f002:**
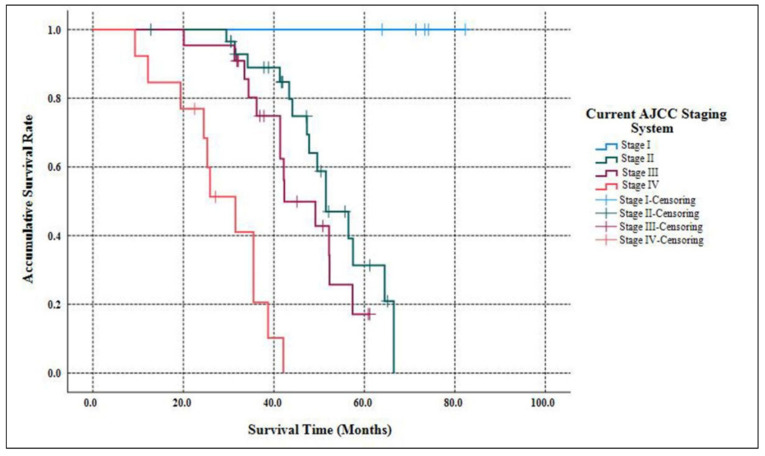
Kaplan–Meier estimates for the OS of G3 p-NETs according to the current AJCC 8th edition staging system originally proposed for G1/G2 p-NETs.

**Figure 3 jcm-11-05253-f003:**
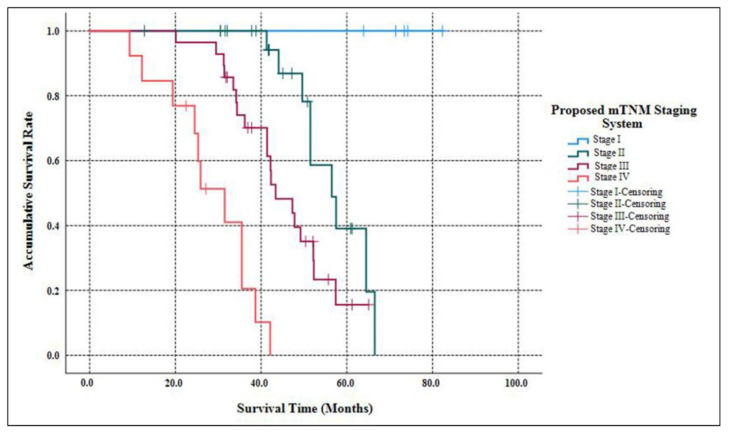
Kaplan–Meier estimates for the OS of G3 p-NETs, according to the newly proposed mTNM staging system primarily designed for G1/G2 p-NETs.

**Table 1 jcm-11-05253-t001:** Clinical features of patients with G3 p-NETs in our study.

Factor	Patients (*n* = 80)	
	No.	%
Patients’ gender, female	50	62.5
Patients’ age at diagnosis, years	50 (7–74)	
Tumor largest diameter, cm	4.5 (1.8–8.5)	
Tumor location, pancreatic body/tail	51	63.8
Tumor number, solitary	71	88.8
Tumor type, non-functional	54	67.5
Incidental diagnosis	15	18.8
Ki-67 proliferation index, %	28 (21–60)	
Mitotic rate, per 10 HPFs	22 (20–38)	
Presence of CgA	80	100
Presence of necrosis	10	12.5
No. lymph nodes harvested (median)	8 (4–14)	
T classification by both TNM systems ^**A**^		
T1	12	15.0
T2	14	17.5
T3	32	40.0
T4	22	27.5
Nodal metastasis (*n* = 24)		
1–3 regional lymph node metastases	15	18.8
≥4 regional lymph node metastases	9	11.3
Distant metastasis	13	16.3
Current AJCC 8th staging system		
Stage I	5	6.3
Stage II	35	43.8
Stage III	27	33.8
Stage IV	13	16.1
Proposed mTNM staging system		
Stage I	9	11.3
Stage II	27	33.8
Stage III	31	38.8
Stage IV	13	16.1

Abbreviations: G: grading; p-NETs: pancreatic neuroendocrine tumors; HPFs: high power fields; CgA: Chromogranin A; TNM: tumor–node–metastasis; AJCC: American Joint Committee on Cancer; mTNM: modified tumor–node–metastasis. ^**A**^: The definitions of T classification in the proposed mTNM staging system were the same as those in the current AJCC 8th staging system.

**Table 2 jcm-11-05253-t002:** Surgical outcomes and follow-up data of patients with G3 p-NETs in our study.

Factor	Patients (*n* = 80)	
	No.	%
Operation classification, surgical resection	62	77.5
Surgical margin (*n* = 62), radical **^A^**	56	90.3
Surgical procedures		
Local resection of pancreatic tumor (enucleation included)	11	13.8
Distal pancreatectomy	28	35.0
Pancreaticoduodenectomy	17	21.3
Radical resection for metastatic disease	6	7.5
Palliative operation with biopsy **^B^**	18	22.5
Anesthesia grade by ASA		
I	9	11.2
II	23	28.8
III	30	37.5
IV	18	22.5
V	0	0
Volume of perioperative blood transfusion (*n* = 24), mL	300 (100–1000)	
Duration of operation, min.	150 (40–340)	
Duration of ICU in-hospital stay (*n* = 15), d.	3 (1–9)	
Duration of postoperative in-hospital stay, d.	6 (3–15)	
Duration of total in-hospital stay, d.	9 (6–20)	
Postoperative complications (*n* = 21)		
Pancreatic fistula	10	12.5
Intra-abdominal infection	7	8.8
Delayed gastric emptying	4	5.0
Intra-abdominal hemorrhage	2	2.5
Reoperation	1	1.3
In-hospital death	0	0
Postoperative medical therapy (*n* = 34)		
Novel molecular targeting treatment	14	17.5
Traditional chemotherapy	20	25
Patient prognosis		
Follow-up time, mon	58.3 (9.7–182.6)	
Out of contact	10	12.5
Dead at follow-up (*n* = 70)	39	55.7
Accumulative 5-year OS	29.7%	
MST, months	49.2 (95% CIs: 41.8–56.5)	

Abbreviations: G: grading; p-NETs: pancreatic neuroendocrine tumors; ASA: American Society of Anesthesiologists; ICU: intensive care unit; OS: overall survival; MST: median survival time. **^A^**: Referring to resections with negative surgical margins, both grossly and microscopically. **^B^**: Referring to cholangiojejunostomy, gastrojejunostomy, etc. with simultaneous biopsy when the local lesion was unresectable or distant metastasis was detected during the intraoperative exploration.

**Table 3 jcm-11-05253-t003:** Univariate and multivariate analyses of prognostic factors for predicting the OS of G3 p-NETs in our study.

Factor	Univariate Analysis		Multivariate Analysis	
	HR (95% CIs)	*p*	HR (95% CIs)	*p*
Patients’ gender				
Male **^A^**				
Female	1.244 (0.864–1.653)	0.236		
Patients’ age				
<50 **^B^**				
≥50	0.931 (0.512–1.349)	0.121		
Tumor location				
Head/uncinate				
Body/tail	1.012 (0.626–1.431)	0.415		
Tumor type				
Functional				
Non-functional1	1.425 (0.712–2.324)	**0.012**	1.034 (0.523–1.517)	0.512 **^C^**
			1.213 (0.671–1.642)	0.214 **^D^**
Incidental diagnosis				
No				
Yes	0.973 (0.5157–1.436)	0.478		
Tumor diameter				
<4.5				
≥4.5	1.479 (0.762–2.962)	**0.016**	0.783 (0.361–1.452)	0.257
			0.981 (0.382–1.901)	0.538
Ki-67 index				
<28				
≥28	2.069 (0.982–4.123)	**0.035**	1.253 (0.564–2.122)	0.873
			0.902 (0.468–2.093)	0.435
Mitotic rate				
<22				
≥22	1.214 (0.614–1.892)	0.125		
Necrosis				
Absent				
Present	3.024 (1.243–7.146)	**0.027**	1.441 (0.684–2.679)	0.137
			0.993 (0.414–1.983)	0.561
Harvested lymph nodes				
<8				
≥8	1.001 (0.425–1.458)	0.512		
Operation classification				
Resection				
Palliative	2.221 (1.329–4.186)	**<0.001**	1.523 (0.723–3.215)	**0.031**
			1.734 (0.757–3.953)	**0.027**
Duration of operation				
<150				
≥150	1.275 (0.546–2.325)	0.343		
Duration of postoperative in-hospital stay				
<6				
≥6	1.241 (0.547–1.874)	0.952		
Postoperative complications				
No				
Yes	0.893 (0.434–2.082)	0.517		
Postoperative medical therapy				
No				
Yes	2.145 (0.783–3855)	**0.042**	1.314 (0.424–2.325)	0.518
			1.211 (0.384–1.924)	0.892
Current AJCC 8th staging system ^E^				
Stage I/II				
Stage III/IV	3.124 (1.322–5.478)	**<0.001**	5.363 (0.329–10.013)	**0.017**
			NA	
Proposed mTNM staging system **^E^**				
Stage I/II				
Stage III/IV	3.954 (0.996–8.326)	**<0.001**	NA	
			3.213 (0.626–8.217)	**0.032**

**^A^**: This related factor was regarded as a reference in the Cox analysis. **^B^**: The value of “median” for quantitative variables was regarded as the cut-off in the Cox analysis. **^C^**: The upper results of the multivariate analysis for each factor were demonstrated in Cox hazard models with the current AJCC 8th staging system. **^D^**: The bellow results of the multivariate analysis for each factor were demonstrated in Cox hazard models with the proposed mTNM staging system. **^E^**: The potential prognostic value of two different systems was demonstrated in separate Cox hazard models. Abbreviations: OS: overall survival; G: grading; p-NETs: pancreatic neuroendocrine tumors; AJCC: American Joint Committee on Cancer; mTNM: modified tumor–node–metastasis; HR: hazard ratio; CIs: confidence interval; NA: not applicable.

## Data Availability

The datasets analyzed during the current study are not publicly available because these materials also form part of an ongoing study, but are available from the corresponding author on reasonable request.

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
