# Peer review of "Surgical Outcomes, Long-Term Survivals and Staging Systems of World Health Organization G3 Pancreatic Neuroendocrine Tumors"

_jcm, 2022, doi:10.3390/jcm11185253_

Round 1
Reviewer 1 Report
This manuscript titled “Surgical outcomes, long-term survivals and staging systems of World Health Organization G3 pancreatic neuroendocrine tumors” is interesting and the authors advocated the importance of mTNM staging system to stratifying G3 pancreatic neuroendocrine tumors and I agree with their opinion. However, there are some minor issues in this manuscript and it is necessary to correct some points. My suggestions are described below;
1) As for analysis of prognostic factors, the number of harvested LN or LN ratio (the number of metastatic LN per the number of retrieved LN) is also important. So, if possible, these factors should be added.
2) Table 3 is confusing and difficult to understand at a glance. So, it should be revised to be more concise.
3) In line 163, there was a typographical error “stagy”, which should be revised.
Reviewer 2 Report
This is an excellent study that explores an area in need of improved diagnostic and prognostic stratification. The methods are sound and the results have been appropriately interpreted. The formatting of Table 3 needs to be improved.
There has been recent interest in the significance of necrosis in pancreatic NET, with a recent paper suggesting that the presence of necrosis in G1/G2 NETs should upgrade them to G3 NETs (PMID: 35934531). The paper would make a more significant and relevant contribution to the literature if it were to address the issue of necrosis in this cohort. It would be interesting to see what proportion of their G3 NETs had necrosis and whether this was a significant prognostic factor. Moreover, it would be really interesting to examine the G1/G2 NETs to look for necrosis and to see if the prognosis of these tumours is comparable to the G3 tumours in this cohort. This latter suggestion may need to be addressed in a follow-up study.
The paper also requires modifications to the English language.
